# Open-source Arduino-compatible data loggers designed for field research

Andrew D. Wickert[1,2,3], Chad T. Sandell[3], Bobby Schulz[1,3,4], and Gene-Hua Crystal Ng[1,2]

[1]Department of Earth Sciences, University of Minnesota, John T. Tate Hall, 116 Church St. SE, Minneapolis, MN 55455, USA.
[2]Saint Anthony Falls Laboratory, University of Minnesota, 2 3rd Ave. SE, Minneapolis, MN 55414, USA.
[3]Northern Widget LLC, Saint Paul, MN, USA.
[4]Department of Electrical and Computer Engineering, University of Minnesota, Kenneth H. Keller Hall, 200 Union St. SE, Minneapolis, MN 55455, USA.

**Correspondence:** Andrew D. Wickert (awickert@umn.edu)

**Abstract.** Automated electronic data loggers revolutionized environmental monitoring by enabling reliable high-frequency measurements. However, the potential to monitor the complex environmental interactions involved in global change has not been fully realized due to the high cost and lack of modularity of commercially available data loggers. Responding to this need, we developed the ALog series of three open-source data loggers, based on the popular and easy-to-program Arduino microcontroller platform. ALog data loggers are low cost, lightweight, and low power; they function between $-30°C$ and $+60°C$, can be powered by readily available alkaline batteries, and can store up to 32 GB of data locally. They are compatible with standard environmental sensors, and the ALog firmware library may be expanded to add additional sensor support. The ALog has measured parameters linked to weather, streamflow, and glacier melt during deployments of days to years at field sites in the USA, Canada, Argentina, and Ecuador. The result of this work is a robust and field-tested open-source data logger that is the direct descendant of dozens of individuals' contributions to the growing open-source electronics movement.

## 1 Introduction

Studies of complex environmental systems require high-density and widespread environmental data (Lovett et al., 2007). Such information is necessary to establish baseline environmental conditions, track global change, and to build theory that is consistent with observations. In spite of three decades of rapid advances in measurement technology (Hirschfeld, 1985; Martinez et al., 2004; Hart and Martinez, 2006; Ferdoush and Li, 2014), most of Earth's surface still needs higher resolution monitoring in order to understand the consequences of global change and prepare for the future (Vitousek, 1994; Tauro et al., 2018). This shortfall results primarily from instrumentation cost (Oliveira and Rodrigues, 2011), hardware requirements to work in harsh environmental conditions, obstacles to advanced technology and repairs in less-developed countries (e.g., Reda et al., 2017), and power and data-retrieval limitations (Martinez et al., 2004; Padhy et al., 2005).

The growing open-source electronics movement has given scientists new tools to develop technologies for both lab and field research (Harnett, 2011; Pearce, 2012; Cressey, 2017), including automated data loggers (Fisher, 2012; Wickert, 2014;

Hicks et al., 2015; Hund et al., 2016; Beddows and Mallon, 2018; Hicks et al., 2019). These innovations have led to significant advances in research and monitoring (Tauro et al., 2018). What the field-monitoring community requires from the open-source movement is a low-power, modular, single-board data logger that is easy to use and whose code and hardware designs are well documented and freely available.

5    We answered this need by developing the "ALog" (Arduino Logger), a small, lightweight, and low-power data logging system that is a fraction of the cost of conventional proprietary data-recording systems (Wickert and Sandell, 2017; Sandell et al., 2018). Hardware advances alone cannot produce an effective standalone measurement platform, so we paired our new designs with custom-built firmware libraries – built atop the popular and easy-to-use Arduino platform – and software to streamline data-logger programming. We iterated development and field testing from 2010 to present, and deployed each 10 round of prototypes across rugged environments, including glaciers, tundra, the high alpine, and wetlands (Wickert, 2014; Tauro et al., 2018; Saberi et al., 2019). Here we present a suite of modern open-source data loggers and the principles that guided their development.

## 2   Design

The ALog series of three data loggers (Figure 1) were designed as an integrated set of hardware, software and firmware 15 (Figure 2). Together, these layers of the embedded system and its interfaces enable low-power data collection. The hardware includes a set of subsystems to manage power, data storage, timekeeping, sensor interfacing, and connections to computers (for testing, clock setting, and programming). This hardware is tightly integrated with the "ALog" firmware library (Wickert et al., 2018a), which manages low-level utilities (power, boot sequence, fail-safes), on-board hardware (the clock and data storage systems, through their own libraries), and a library of sensor commands. The full system is programmable through the 20 Arduino integrated development environment (IDE), which is designed for use by beginner programmers and therefore lowers the barrier to entry for environmental monitoring. All hardware designs and code for the fully open-source ALog system are available at https://github.com/NorthernWidget. Core firmware and software are licensed under the GNU General Public License (GPL), which requires that all derivatives of the ALog remain open-source, and hardware is released under the Creative Commons Attribution Share-Alike license.

25    To design the ALog series of data loggers, we followed the approach taken by the popular Arduino project (Barragán, 2004; Banzi and Shiloh, 2014) in order to maintain compatibility with open-source standards. We designed the circuitry (hardware) using EAGLE (Cadsoft and Autodesk, 2019), an electronic schematic and board-layout program that is freely available for non-commercial use and is a *de facto* standard across the open-source community. We wrote the data logger software in the Wiring/Arduino variant of C++ (Barragán, 2004), using an object-oriented framework that abstracts the low-level core compo- 30 nents of embedded hardware programming (e.g., writing bits to registers) into intuitive functions to read, write, and operate a microcontroller. The programmable core of the ALog is compatible with Arduino, enabling the use of its extensive firmware libraries and IDE. Multiple examples are bundled with the ALog library, which is fully documented using doxygen (van Heesch, 2008), a program that automatically converts in-code documentation into a users' manual. We customized doxygen to include

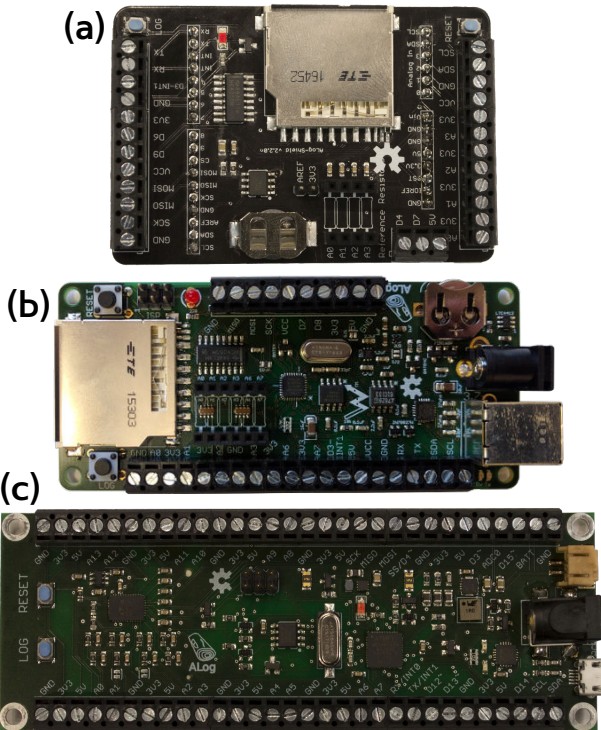

**Figure 1.** Photos of the ALog series of data loggers. **(a)** The ALog Shield 2.2 (Wickert et al., 2018b), which nests atop a standard Arduino board. **(b)** The ALog BottleLogger 2.2 (Wickert and Sandell, 2017). **(c)** The ALog BottleLogger 3.0 (Sandell et al., 2018); the battery and SD card holders are on the reverse side.

the README Markdown file, which includes instructional text and images, at the beginning of the automatically generated manual. The customized doxygen configuration is available at https://github.com/NorthernWidget/ALog/tree/master/doc (Wickert et al., 2018a), and generated reference manual is included in the Supplement).

## 2.1 Hardware

5 The ALog series comprises three main data loggers (Figure 1): (a) the ALog Shield 2.2 (Wickert et al., 2018b), (b) the ALog BottleLogger 2.2 (Wickert and Sandell, 2017), and (c) ALog BottleLogger 3.0 (Sandell et al., 2018) (Table 1). The ALog BottleLoggers are standalone units (i.e., no built-in telemetry) that we designed for field research; version 3.0 has a more powerful microcontroller core and a dedicated 16-bit analog-to-digital converter (ADC), whereas version 2.2 uses less power and fewer components. Both are described in more detail in this section. The ALog Shield 2.2 functions as an entry point for

10 the global community of Arduino users (Buechley and Eisenberg, 2008; Cressey, 2017) to develop their own scientific data logging capabilities. It nests atop a standard Arduino board (Barragán, 2004; Banzi and Shiloh, 2014), such as the Arduino Uno. The power consumption of this pair is too high for field deployment, but together these form a benchtop prototyping

**Table 1.** ALog attributes.

| | Shield 2.2 + Uno | BottleLogger 2.2 | BottleLogger 3.0 |
|---|---|---|---|
| Width [mm] | 78.7 | 44.1 | 44.4 |
| Length$^a$ [mm] | 85.7 | 113 | 120.8 |
| Mass$^a$ [g] | 63.49 | 42.66 | 53.84 |
| Input voltage [V] | 3.3–12 | 3.5–5.0 | 2.5–12 |
| Power (sleep)$^b$ [$\mu$A] | 34000 | 12 | 80 |
| Power (awake)$^b$ [mA] | 54 | 7.5 | 11.9 |
| MCU | ATMega 328p | ATMega 328p | ATMega 644p$^c$ |
| Clock speed [MHz] | 16 | 8 | 8 |
| Program memory [KB] | 32 | 32 | 64 |
| Variable memory [KB] | 2 | 2 | 4 |
| EEPROM [KB] | 1 | 1 | 2 |
| External Interrupts | 1 | 1 | 2 |
| Analog I/O | 4×10-bit | 6×10-bit | 16×16-bit |
| Digital I/O$^d$ | 4 | 2 | 6 |
| I$^2$C | 1 | 1 | 2 |
| SPI | 1 | 1 | 1 |
| Dedicated UART | 0 | 0 | 1 |
| Data storage [GB] | 32 | 32 | 32 |
| RTC drift [s/day] | ±0.432 | ±0.432 | ±0.432 |

$^a$ Including the SD card and backup battery

$^b$ At 4.5V input; "awake" state is not including additional power draw from sensors

$^c$ Also compatible with the ATMega1284p, with 128 KB program memory, 8 KB variable memory, and 4 KB EEPROM.

$^d$ 10-bit analog I/O also functions as digital I/O.

system that is compatible with the ALog firmware (Wickert et al., 2018a). The Supplement contains the electrical schematics and circuit board layouts for all three data loggers.

Each data logger of the ALog series contains six critical subsystems (Figure 2): power, timekeeping, data storage, sensor interfaces, input/output (I/O), and the microcontroller core. The high-efficiency power system permits multi-year deployments on a single set of primary alkaline batteries. These battery lifetime measurements are based both on extrapolation from laboratory power-consumption measurements (Table 1) and field deployments that remained unvisited for $\geq 1$ year (Armstrong et al., 2016). A high-accuracy real-time clock (RTC) keeps time, regulates logging intervals, and is temperature-compensated to reduce drift to ±0.432 seconds per day (firmware implementation by Ayars and Wickert, 2018). Data are written as ASCII comma-separated text files to Secure Digital (SD) cards for low-cost, high-volume storage. Screw terminals connect sensors

and other peripherals to the ALog, where they link to the appropriate interfaces on the microcontroller. The ALog communicates with and is programmed by a computer via a USB–serial converter that links the computer's USB interface with the universal asynchronous receiver-transmitter (UART) of the microcontroller. Each ALog is built around a reprogrammable 8-bit AVR microcontroller (Wollan et al., 1998; Bogen and Wollan, 1999).

While it is a simple design decision, using an SD card instead of internal memory has multiple advantages. The open-source "SdFat" interface library written by Greiman (2016) allows up to 32 Gb of data to be written in human-readable comma-separated ASCII format. By recording data in text files on SD cards, we also simplify data download and visualization, making it easier for field staff and citizen scientists to work with the ALog. Removable storage limits the time that the box must be opened and exposed to the elements, and also allows SD cards to be swapped in the field. We chose standard large-format SD

cards because they include space to write physical notes and because smaller cards are more easily lost in the field.

We greatly reduced power consumption – a key system feature – while simplifying power supply options. To reduce power consumption, we implemented a "sleep" cycle to shut down all non-essential subsystems while not logging, utilized a lower-speed (8 MHz) crystal to set the processor clock speed, and powered the ALog using either an unregulated power supply (BottleLogger 2.2) or a step-up–step-down (buck–boost) converter with a high-efficiency switching architecture (BottleLogger

3.0). Power may be supplied by primary alkaline cells – commonly available across the globe – or through a solar panel, rechargeable battery (typically lithium-ion), and charge controller. When powered by 3 in-series AA alkaline cells (∼2600 mAh for these calculations) and awake (logging) for one second per minute, the ALog BottleLogger 2.2 can run for ∼2 years and the BottleLogger 3.0 can for ∼1 year, based on our laboratory measurements of power consumption by the data logger alone. This time may decrease if sensors that require significant power compared to the "awake" state of the logger are attached

(Table 1). If this is the case, D cells (>10,000 mAh) may be a suitable alternative. In our field deployments (Section 3), we typically recorded data once every ten minutes, further increasing battery life. As a result of this low power consumption, we ran our ALog field deployments exclusively with primary alkaline cells. In addition, any ALog may be powered over USB, and a diode array prevents short circuits between USB and external power supplies.

In order to reduce power consumption, which is especially important for remote field deployments, we decided not to include

on-board telemetry. Off-board radio (e.g., RFM95, XBee), mobile phone (e.g., Particle Electron, Particle Boron), or satellite (SPOT, Iridium) telemetry packages could be added through the exposed digital interfaces on the ALog data logger. However, such additions would require their own significant power paths, including rechargeable batteries, charge controllers, and solar panels, thus negating much of the low-power benefit of the ALog BottleLogger design. In contrast to the ALog BottleLogger, the open-source Mayfly data logger (Hicks et al., 2015, 2019) includes an XBee radio header (Hicks et al., 2019) and firmware

support for radio telemetry (Aufdenkampe et al., 2017; Damiano et al., 2019). In addition, Adafruit ("Feather") or Particle Internet-of-Things ("IoT") boards can serve as low-cost platforms for telemetry; these do not necessarily include an on-board RTC or data storage, reducing cost but making consistent telemetry critical. These alternatives to the ALog BottleLogger are effective options for deployments in which data return and not power consumption is the variable to optimize.

ALog data loggers are designed for versatility in the field. Each ALog is only ∼50 g due to the low mass of solid-state

electronic components mounted on a circuit board (Table 1). Their low mass and associated small size, along with the stability

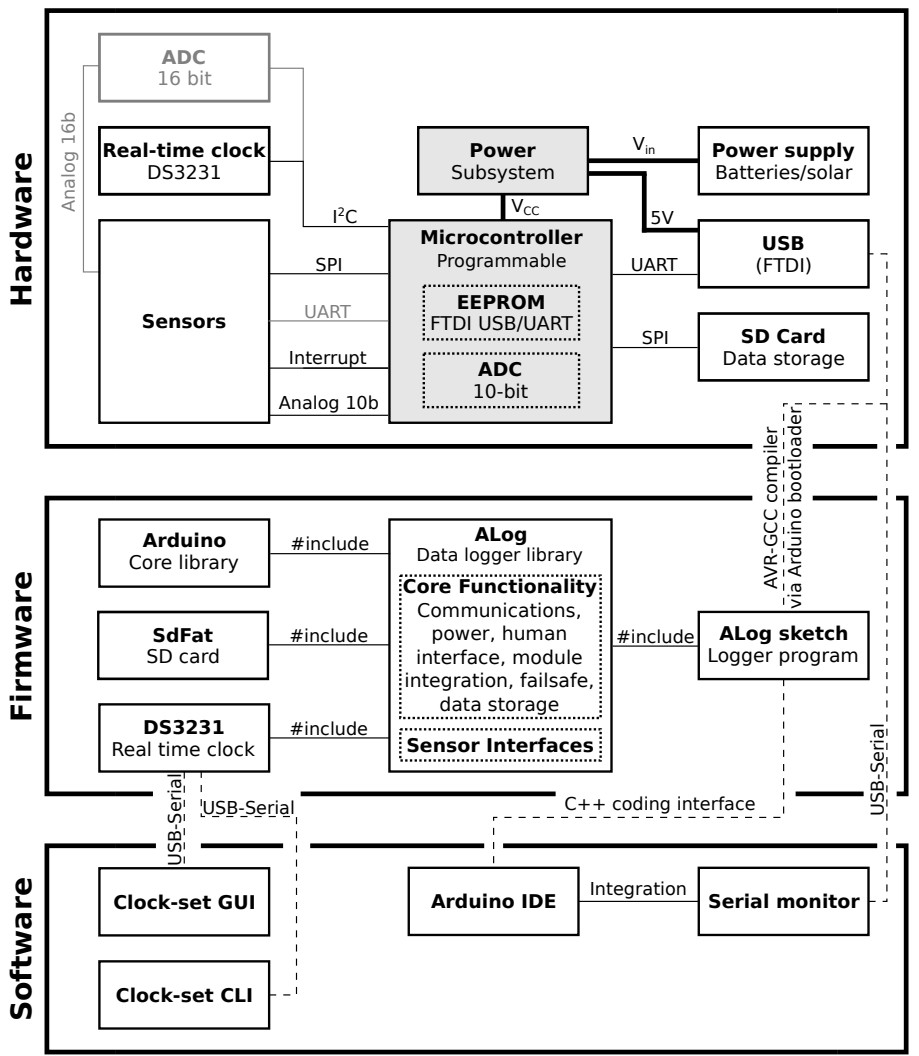

**Figure 2.** Flowchart of ALog hardware, firmware, and software. "Hardware" includes the generalized subsystems of the physical ALog. "Firmware" includes all of the code that runs on the AVR microcontroller. "Software" runs on the user's computer. Dashed lines indicate temporary connections (e.g., during programming), whereas solid lines indicate permanent connections and dependencies. Dotted lines within boxes contain information on components of larger systems. Lines in gray indicate features that are included only on the BottleLogger 3.0; gray-shaded boxes denote features that are incorporated into the BottleLogger designs but for the ALog Shield are supplied by a standard Arduino. ADC: analog–digital converter; EEPROM: non-volatile variable memory that holds values after a power reset; $I^2C$, SPI, UART, interrupt, analog: communications protocols; GUI: graphical user interface; CLI: command-line interface.

**Table 2.** ALog labor and pricing. All costs are given in US dollars. Component costs were found at Digi-Key (https://www.digikey.com/). Board prices were sourced from OSH Park (https://oshpark.com/). Build and testing times are based on our work with the data loggers, and the labor costs are estimated based on quotes from Caltronics Design & Assembly (https://caltronicsdesign.com/). All values are per board, and all prices were determined on 12 January 2019.

|  | Shield 2.2 | BottleLogger 2.2 | BottleLogger 3.0 |
| --- | --- | --- | --- |
| Components (QTY 1) | $22.24 | $43.38 | $59.46 |
| Components (QTY 100) | $14.45 | $28.30 | $39.43 |
| Board (QTY 3) | $10.87 | $11.28 | $13.85 |
| Board (QTY 100) | $6.52 | $6.77 | $8.31 |
| Build & test time (QTY 1) | 20 min. | 75 min. | 120 min. |
| Build & test labor est. (QTY 100) | $20 | $35 | $45 |

of solid-state electronics, help users to carry ALog data loggers into the field and reduce the chance of damage if they are dropped. All onboard electronics are rated to function between $-40°C$ and $+85°C$ (standard "industrial" components); commonly available batteries can power the ALog between temperatures of $-30°C$ and $+60°C$. ALog data loggers are inexpensive and accessible because they are built from standard off-the-shelf components that are available in most of the world. Further reducing total system cost and increasing versatility, the ALog's generalized set of sensor interfaces allows it to read data from common and inexpensive commercially available sensors. This sets it apart from closed-source data loggers, which are designed to interface with specific proprietary sensors. This combination of affordability and accessibility can help to expand the reach of automated environmental observations and reduce the financial risk associated with recording data in hazardous or unsecured locations. Finally, the open-source schematics and circuit board layout assist users in diagnosing and repairing their own ALog data-logger systems.

## 2.2 Firmware

We built a firmware library (see supplementary design files) that streamlines ALog programming through a modular two-component architecture (Wickert et al., 2018a). The first is a set of utilities that manage logger core functionality. The second is a library of functions that communicate with and record data from sensors (Table 3). This separation prevents users from altering the code that manages core logger functions when adding or editing sensor functions. We classify this code as "firmware" rather than "software" because it is uploaded to the microcontroller as a semi-permanent set of instructions that exists in program memory until it is externally wiped and replaced. The ALog library is written in in the Wiring/Arduino variant of C++, which is the standard for open-source microcontrollers (Barragán, 2004; Banzi and Shiloh, 2014). It maintains compatibility with any standard Arduino-compatible device in order to ensure that the ALog firmware can support the open-source community even in the absence of ALog hardware. This library is then imported, and its core "ALog" class instantiated, within an ALog program (Arduino "sketch") that is uploaded to the data logger.

**Table 3.** Measurements and sensors currently supported by the ALog. Support for additional sensors may be added to the open-source ALog firmware library (Wickert et al., 2018a), following the _sensor_function_template example, which may be found in the design files and reference manual in the Supplement.

| Property | Sensor | Communications |
|---|---|---|
| Temperature | Thermistor | Analog R |
| Temperature | BMP280 | Analog R |
| Soil moisture | Dielectric probe | Analog V, UART |
| Rainfall | Tipping-bucket rain gauge | Interrupt |
| Wind speed | Cup anemometer | Interrupt |
| Wind direction | Magnetic wind vane | Analog R |
| Distance | Ultrasonic rangefinder | Analog V, UART |
| Distance | Linear potentiometer | Analog R |
| Absolute pressure (atmos.) | Digital barometer (BMP280) | $I^2C$ |
| Absolute pressure (water) | Sealed pressure transducer | Analog V, $I^2C$ |
| Solar radiation | Pyranometer with amp. | Analog V |
| Relative humidity | Humidity probe | Analog V |
| Image or video | Camera trigger | Digital I/O |
| Groundwater temp. and flux | Thermal profiler | Analog R ($\times 6$) |
| Overland flow status | Binary conductivity sensor | Analog R |
| Angle or tilt | 2-axis inclinometer | Analog V ($\times 2$) |
| Force | Force-sensitive resistor | Analog R |

The core-utilities portion of the library manages its power and logging cycle, and interfaces with the user, SD card, and RTC. The firmware reduces power consumption by a factor of 150–625 from an "always-on" state (Table 1) by instructing the system to spend most of its time in a low-power "sleep" mode in which all non-essential systems are either shut down or put into low-power modes themselves. Errors during the logging cycle are caught by the watchdog timer, which reboots the ALog if it hangs and writes a time stamp to a file to help in uncovering the source of the error. The user interface comprises status and sensor messages passed to the serial monitor as well as a set of status messages flashed by the LED.

The sensors component includes both private utility functions for standard operations (e.g., calculating standard deviations, solving the voltage-divider equation) and public functions that link each analog or digital sensor interface to the ALog and record data. Sensor functions are modular and written following a standard inputs–processing–outputs template. Outputs are written to the SD card as plain text ASCII data and header files, and printed to a serial monitor if the data logger is connected to a computer. Current support exists for a broad range of off-the-shelf sensors (Table 3), many of which are inexpensive. Users may add additional sensor support to the ALog library with help from a function template (design files in the Supplement) and the documentation (reference manual in the Supplement). Users may then contribute their code for additional sensor support to the main ALog repository, thereby increasing the reach of open-source instrumentation.

Sensors may be read on a standard interval or in response to an event. When reading measurements at a standard interval (typically 1–10 minutes), the RTC wakes the ALog using an interrupt. Once awake, the ALog retrieves data from all sensors, which usually measure environmental states. Reading and recording data from these sensors typically takes 1–3 seconds, during which the ALog is operating in its high-power "awake" state (Table 1). An event-based impulse, such as that from a tipping-bucket rain gauge, instantaneously wakes the ALog, which then records a time stamp to a different data file from that which is used for regular RTC-driven measurements. Reading and recording this time stamp typically requires <0.3 s of awake-state power consumption. If the ALog is already awake (e.g., during RTC-driven data logging) when an event occurs, the ALog firmware records the time of the event to its file and then continues the remainder of its ongoing task.

To program ALog data loggers, users import the ALog library into an Arduino sketch and instantiate the ALog class. Using examples included with the ALog library as a guide, users write a set of instructions that prescribe which sensors should be read and how often data should be recorded (Figure 3). This sketch is then compiled and uploaded to the ALog as firmware.

## 2.3  Software

Arduino sketches to program the ALog may be written and uploaded using the Arduino IDE (Banzi and Shiloh, 2014), which evolved from Processing (Reas and Fry, 2007) and Wiring (Barragán, 2004). The Arduino IDE contains an interface to automatically download and install the custom ALog hardware definitions files (including the appropriate bootloaders) and code libraries. It also includes a serial monitor to view communications between the ALog and the computer.

The ALog clock is set via the USB–serial connection using a digital handshake programmed within the ALog library. Two options to set the clock are available: a command-line serial interface program written in Python (Wickert, 2017) and a graphical program written in Processing (Schulz, 2018) (see supplementary design files). Both methods interact with the ALog immediately upon boot-up (see Figure 3).

## 3  Field deployment

ALog development evolved iteratively over more than eight years of lab design and field deployments (Figure 4). ALog data loggers have been deployed in the high alpine (Niwot Ridge, Colorado, USA), the high desert (Quebrada del Toro, Salta, Argentina), coastal wetlands (Wax Lake Delta, Louisiana, USA), subalpine valleys (Gordon Gulch, Colorado, USA), tropical mountains (Volcán Chimborazo, Ecuador), continental lacustrine regions (Minnesota, USA, and Ontario, Canada), and on a large valley glacier (Kennicott Glacier, Alaska, USA) (Wickert, 2014; Armstrong et al., 2016; Tauro et al., 2018; Saberi et al., 2019). Field deployments ranged from a few days to three years. During these deployments, the ALog recorded data from weather stations, glacier-ablation monitoring stations, thermistors, stream gauges, soil moisture probes, pressure transducers for water levels in wells, subsurface temperature profilers, and frost-heave gauges; Table 3 contains a full list of sensors for which firmware has been developed and included in the ALog library (Wickert et al., 2018a). These field deployments tested the ALog data logger in humid and dry environments, onshore and offshore, and in temperatures that ranged from $-30°C$ to $+35°C$.

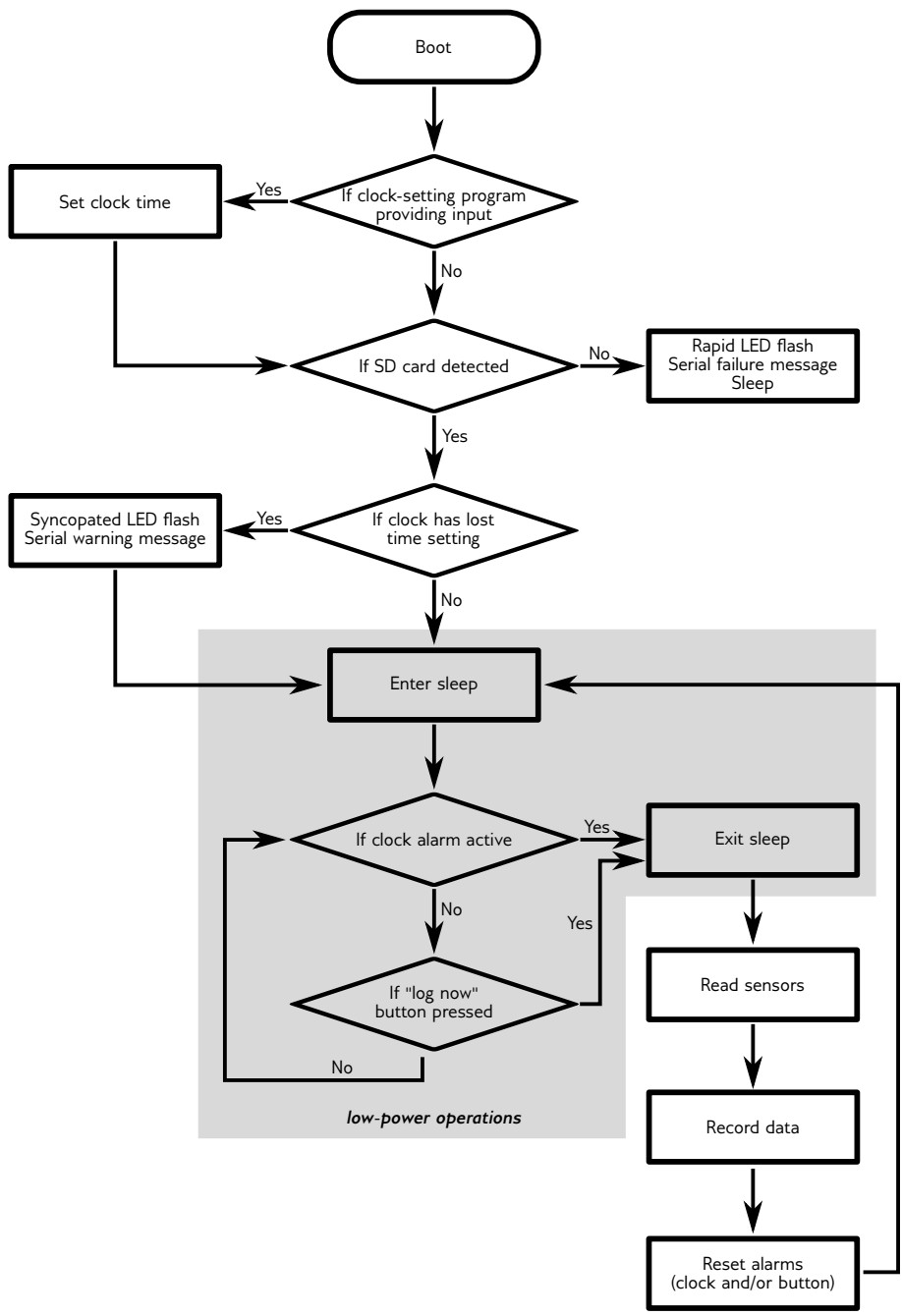

**Figure 3.** Flowchart of ALog operations. This set of steps is prescribed by the firmware. Not pictured is the watchdog timer, which resets the data logger, returning it to the "boot" step, if it hangs for more than 8 seconds.

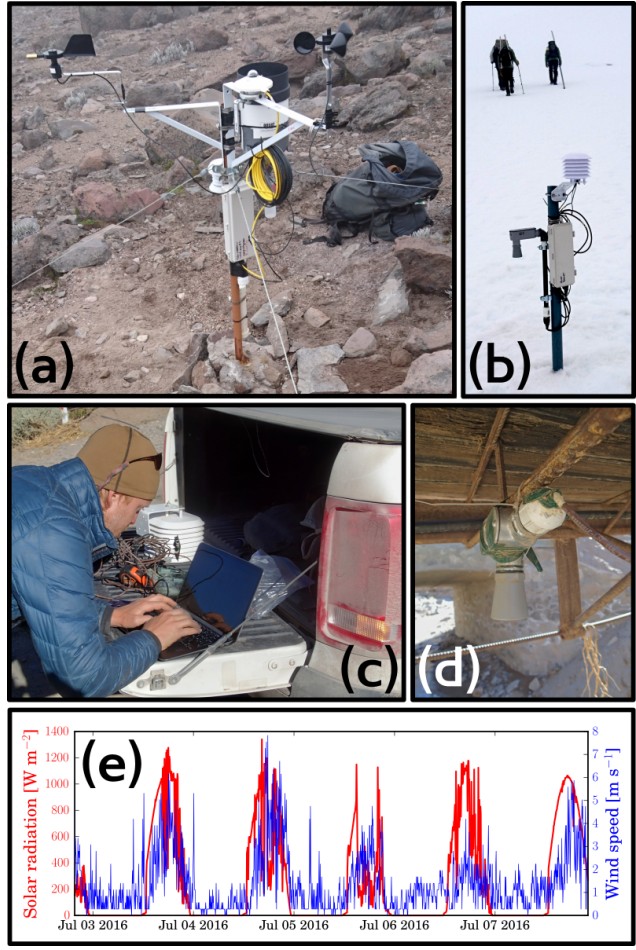

**Figure 4.** Example ALog deployments. **(a)** Weather station with anemometer, pyranometer, wind vane, thermistor, and tipping-bucket rain gage, to measure high-desert climate parameters. **(b)** Glacier monitoring station with look-down ultrasonic sensor, thermistor, and relative humidity sensor, to monitor ablation and its drivers. **(c)** Downloading field data by copying and pasting an ASCII text file from the SD card. **(d)** Look-down ultrasonic sensor as a simple stream gauge; despite the destruction of the monitoring system in a historic flood event, the data file remained saved and uncorrupted on the SD card. **(e)** Solar radiation and wind speed covary at the station in (a); wind speed lags radiation, indicative of surface-heating-driven convective winds in this high-altitude arid environment.

The final ALog designs were guided as much by failure as by success. When the ALog failed in the field, it was typically due to (1) moisture intrusion, (2) failure to properly seat the SD card, (3) loss of power to the real-time clock that caused it to reset its date and time to midnight on the morning of 01 January, 2000, or (4) poorly written firmware. Moisture intrusion was managed by improving the enclosures (see below). To ensure that the SD card was seated properly, we established a protocol of pressing the reset button upon reinsertion and waiting for a "long–short–short" flash of the indicator LED (see the reference manual in the Supplement). Real-time clock failures are denoted by a syncopated flash on the indicator LED. This flash pattern notifies the user to set the RTC, and the ALogTalk software (Wickert, 2017) records a set of five measurements of the computer time and logger time before setting the RTC. These linked time stamps help the user to manually correct the timestamps on data that were recorded after the clock reset to its factory-default time of midnight on 01 January 2000. To recover from firmware errors that could cause failures over times long enough that we would not observe them in the lab, we enabled a watchdog timer to reset the ALog if it hung for 8 seconds. On each watchdog-timer reset, the logger writes a time stamp to a file in order to help with data QA/QC and to assist in future debugging.

## 3.1 Enclosures

Choosing an appropriate enclosure is a key decision for equipment survival in the field. The "BottleLogger" moniker comes from its designed form factor that allows it to fit inside a wide-mouth Nalgene bottle. Such bottles seal well and are commonly available from suppliers of both laboratory and outdoor equipment. This design feature was created as an option for occasions when easier-to-use but harder-to-souce enclosures were not available. In the majority of our deployments, we have used acrylonitrile butadiene styrene (ABS) plastic enclosures (models NBF-32104 and NBF-32108). These boxes are gasketed, include lever-style clips for easy opening and closing in the field, are large enough for either 3×D or 3×AA cells, and may be easily drilled or machined to accommodate cable glands for connections to sensors. The shorter enclosure (NBF-32104) requires a right-angle barrel jack plug in order to fit the length of the logger, SD card, and power connector.

We typically attach the loggers to the lid of the enclosure and the battery pack to the bottom of the enclosure using self-adhesive hook-and-loop. This holds both in place, but allows either to be easily removed for wiring. The longer boxes (NBF-32108) permit cable glands to be drilled in the lid next to the logger, reducing the need for cable strain relief. The shorter boxes (NBF-32104) may include cable glands in one or more sides, and this is easier if they are fitted with a 3×AA cell pack (as opposed to a 3×D cell pack).

## 3.2 Examples

We highlight two hydrologically relevant example deployments performed at Volcán Chimborazo, Ecuador, following on the work of La Frenierre (2014) and La Frenierre and Mark (2017). In the first deployment, we measure weather conditions; in the second, we measure glacier ablation and its drivers. In both cases, the ability of the ALog to communicate with multiple sensors from different manufacturers allows these stations to record relevant data to better understand mountain hydrology in the glacierized Andes.

On the arid eastern side of Volcán Chimborazo, we installed an ALog BottleLogger connected to sensors for wind speed (Inspeed Vortex anemometer), wind direction (Inspeed e-Vane), and solar radiation (Kipp and Zonen CMP3 pyranometer linked with our in-house-designed instrumentation amplifier). We affixed these sensors and the ALog BottleLogger to an existing structure that was used to measure rainfall and temperature using a proprietary data-logging system (Figure 4a) (La Frenierre, 2014; La Frenierre and Mark, 2017). Figure 4e contains the first five days of data from this deployment, which lasted one year.

We installed a prototype automated ablation stake on Reschreiter Glacier on the more humid eastern flank of Volcán Chimborazo (Figure 4b). We designed this automated stake to measure both the atmospheric factors that drive snow and ice ablation and the amount of snow and/or ice melt that occurs. Atmospheric variables include temperature, measured using a CanTherm epoxy-embedded thermistor paired with a reference resistor, and humidity, measured with a TE Connectivity HM1500LF sensor; both of these sit within a solar radiation shield (Figure 5b). Distance to the snow and ice surface is measured using a MaxBotix ultrasonic rangefinder (Figure 5a) (see Wickert, 2014), which is paired with a digital inclinometer to check and correct for station tilt as the ablation stake gradually melts out of the snow and/or ice (Figure 5a). This station, here programmed to record data every five minutes, dramatically increases ablation data density beyond traditional methods, which incorporate daily to weekly field surveys of snow and/or ice surface elevation change around ablation stakes. Furthermore, by including on-stake temperature measurements, we are able to compute at-stake melt factors for degree-day melt models, which are significant for both glaciological and water-resources research (Saberi et al., 2019).

## 4 Discussion

The paradigm of global change research has been one of scientists studying, reporting, predicting, and communicating how human activities impact the environment (Syvitski et al., 2009; Foley et al., 2011; Pelletier et al., 2015; Tauro et al., 2018), ideally followed by the broader public responding with plans to better manage Earth's environment and natural resources. In order to develop the ALog, we reversed this flow of information by drawing on open-source hardware, firmware, and software designs from the public to develop a scientific tool (Cressey, 2017). The open-source electronics movement has grown rapidly as part of the "maker revolution", in which individuals develop new technology and share their designs (Buechley et al., 2008; Anderson, 2012; Libow Martinez and Stager, 2013; Hut et al., 2016). We predict that building atop a broad and popular base platform will increase public accessibility to and interest in scientific measurements, and improve the support for and longevity of the data logger technology.

The ALog is part of a community of open-source tools for scientific research (Harnett, 2011; Pearce, 2012; Cressey, 2017) that includes both sensors (Keeler and Brugger, 2012; Barnard et al., 2014; Fatehnia et al., 2016; Hut et al., 2016) and automated data loggers (Fisher, 2012; Hund et al., 2016; Beddows and Mallon, 2018). Of these, the ALog BottleLogger designs have the lowest power consumption (Table 1). Their screw terminals easily communicate with any sensor via multiple methods of analog and digital communication, and their fully integrated and documented firmware (Figure 2) reduces end-user coding to a few lines. These features have been included individually in other firmware and hardware designs (Hund et al., 2016; Aufdenkampe et al., 2017; Damiano et al., 2019), but the ALog incorporates all of these into a single streamlined system.

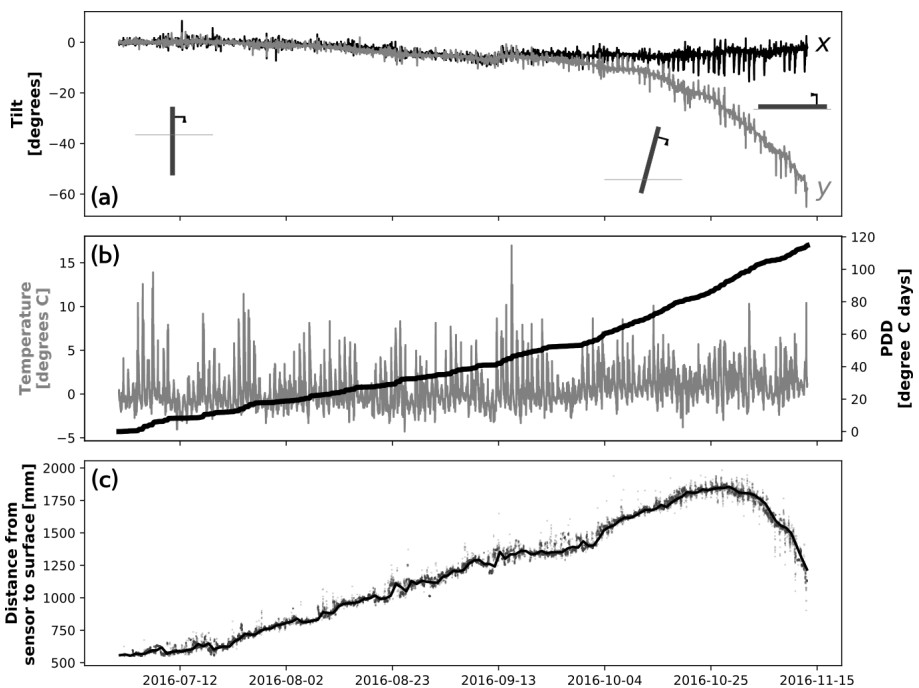

**Figure 5.** Automated ablation stake data. Figure 4b shows an example of an identical ablation stake deployed elsewhere on Volcán Chimborazo (Saberi et al., 2019). **(a)** Inclinometer output as the snow and ice melt, eventually causing the ablation stake to tilt and fall over as pictured in the cartoon drawings. **(b)** Temperature (gray) and cumulative positive-degree days (black). **(c)** Vertical distance from the ultrasonic rangefinder to the surface as single-time measurements (gray semi-transparent points) and a daily moving average (black line). The roll-over indicates the point at which the ablation stake tilt begins to dominate the signal, and until this point, ablation (i.e., increasing distance) generally tracks the positive-degree-day line.

All hardware schematics and software are regularly updated and available from GitHub in formats that can be read by free software. These design decisions are essential to building a broad and active user base that can work collectively to increase environmental monitoring across the globe using high-quality open-source technologies.

## 5   Conclusions

We developed the ALog system of open-source data-logging hardware, firmware, and software; built compatibility for a wide range of sensors; and tested the designs extensively in the field. Multiple design iterations over more than eight years of development decreased power consumption, improved field usability, and led to a simple library-based system to streamline programming into one-line function calls. We created the ALog with the help of community-developed open-source code and designs, and we hope that the ALog in turn can be a stepping stone to even more advanced, usable, and powerful open-source technology. By expanding the reach of open-source field instrumentation, we hope to help scientists and members of the broader community measure and understand our changing world.

*Code and data availability.* All ALog hardware schematics and board layouts, firmware, and software are included in the Supplement. Links to release-version archives and their associated DOIs are provided as assets and referenced herein. Also included in the Supplement are a reference manual and PDF files of the hardware schematics. GitHub hosts up-to-date versions of the data logger hardware (https://github.com/NorthernWidget/ALog-BottleLogger – including the reference manual – and https://github.com/NorthernWidget/ALog-Shield), firmware (https://github.com/NorthernWidget/ALog, https://github.com/NorthernWidget/DS3231, and https://github.com/greiman/SdFat, and clock-setting software (https://github.com/NorthernWidget/ALogTalk and https://github.com/NorthernWidget/SetTime_GUI.

*Author contributions.* ADW conceivced of, designed, prototyped, and developed the ALog BottleLogger and Shield from 2011 to present. CTS updated the ALog BottleLogger v2 design and developed the ALog BottleLogger v3 with assistance from BS. GCN, CTS, and ADW tested and deployed the ALog BottleLogger. ADW wrote the manuscript, GCN and BS edited the manuscript, and CTS provided input.

*Competing interests.* A. D. Wickert, C. T. Sandell, and B. Schulz are members of the company Northern Widget LLC, which develops and distributes the open-source ALog series of data loggers.

*Acknowledgements.* The generosity of the open-source electronics community made this project possible. Design assistance and inspiration were drawn from G. Oberforcher, K. M. Wickert, K. R. Barnhart, S. Hicks, and materials made available by SparkFun Electronics and Adafruit Industries. R. S. and S. P. Anderson, N. Rock, Z. Frederick, and L. G. Dandrea provided motivation and/or field perspectives. Significant field deployments, constructive feedback, and assistance were provided by R. S. Anderson, N. Rock, W. H. Armstrong, S. Rathburn, C. Paola, N. Evans, D. Ward, C. Bode, I. Overeem, P. Nelson, D. Brogan, M. Farin, R. McLaughlin, A. Yourd, J. La Frenierre, B. Putnam, and C. Burnett. ADW was supported by the US Department of Defense through the National Defense Science & Engineering Graduate Fellowship Program, the US National Science Foundation Graduate Research Fellowship under Grant No. DGE 1144083, and a Postgraduate Research Grant from the British Society for Geomorphology (Wiley-Blackwell). Funding awarded to ADW and GCN by the University of Minnesota helped to support this work from 2015–2018. Comments from Rolf Hut and one anonymous referee improved the clarity and scope of this manuscript.

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
