# Peer review of "Open-source Arduino-compatible data loggers designed for field research"

_Hydrology and Earth System Sciences, 2018_

## Referee Comment (RC1) · Hut (Referee) · 11 Jan 2019

The authors present their work on designing an open source data logger. The device presented is, in my opinion, valuable to the hydrological community and the article is written clearly and concise. Apart from a few minor comments that I have listed below, I recommend publishing this work in HESS. Minor comments, numbered for easy replying by the authors:

1. A logger keeps the recorded data "on site". For many applications, among others for operational hydrological services, (near) real time data is needed that is transmitted from the field to some (online) service. I ask the authors to add a few sentences in the introduction to acknowledge this difference and position their

device within the larger collection of existing communication and logger solutions

2. The article focusses on the PCB, but for this to work in the field the choice of enclosure is essential. The name "bottleLogger" hints that a bottle could be used as enclosure, but the authors never make this explicit. Recognizing that the choice of enclosure is very depended on field conditions and choice of sensors, I would still ask the authors to spend a few words on the enclosures that work well (or advice from their experience on what doesn't work!) to help the hydrologists that want to use their device in the field.

3. I applaud the authors for going full "Open Science" and making both their hardware and software available to the community. To make sure that the software version that is part of this publication remains available, even when a new version is published on github (or, heaven forbid, Microsoft decides to shut github down. . .) I ask the authors to

    (a) Make a "release" of their software on github, giving it a version number
    (b) Use a service like Zenodo to get a DOI (and a guaranteed archive) for the software and instead of providing the url in the article, provide a citation to the archived version. (Zenodo facilitates this)

4. The authors indicate that trigger-based sensors (tiping buckets) are supported through interrupts. Please elaborate if and how interrupts work together with the sleep function and how interrupts effect the battery life.

5. One of the main points the authors make is that their Alog is low cost ("a fraction of the cost of conventional proprietary systems"), yet they do not mention any price point or even price range. I ask the authors to indicate what it would cost, at current price points, if a hydrologist use their openly provided designs to produce an Alog (or a batch of them) and what it would cost if they buy if directly from a sales agent.

[Figure]

Disclaimer on language: I'm not a native English speaker and do not check for typos or incorrect spelling or grammar. I review on the content the authors are presenting.

---

## Referee Comment (RC2) · Anonymous Referee #2 · 13 Jan 2019

General comments:

This article describes the development and technical details of the ALog data logger series, an open-source and low cost data logger that is based on Arduino technology. The article and the described data loggers are a significant contribution to the science community and readers of HESS, as the data loggers may provide a useful technology to many environmental scientists. The article summarizes the substantial development that has gone into the data logger development over many years, and provides detailed background information. The article also includes supplemental material and codes provided online. This helps to make the data logger accessible to the science (and general public) community. The article is well written and well organized throughout, with clear descriptions of the technology. I only have a few minor questions, mostly

regarding field deployment, and minor comments that should be addressed. Apart from these, I recommend this article for publication.

Specific Comments:

1) Please also discuss some of the challenges that you have faced with the ALog data logger in the field, and that a potential user of this data logger may encounter and should be aware of. You mention several field experiments with ALog data loggers in adverse conditions. How long were the data loggers actually in the field, how robust were they found to be? What field issues did you encounter that were specific to the ALog? Were you able to remidify these in the next iterations? I realize the ALogs have been developed over a long period, but a few more examples would be helpful to a potential user of the technology.

2) For instance, did you encounter clock drift? It is referred to a really low clock drift value in the article, but was this value based on 'theoretical lab experiments', or tested in the field?

3) The low power use is impressive. Was this value also experienced in the field? I.e., one data logger actually ran for $\sim$ 2 years on three AA batteries? Or is this a theoretical value based on consumption? Also, what kind of sleep-awake cycling is typically used? A 1 second per minute interval is mentioned, was that typically used? It probably depends on sensor and application, but some examples would be good.

4) In line with the above questions, please also include some more information on how the data loggers were installed in the field. What kind of encasing have you found to work well with these data loggers? Do you typically use batteries or solar panels?

Technical Corrections

Line 4 –Alog series: add reference that three data loggers were developed as part of the series.
Line 17 – likely also capacity challenges, especially in developing countries.

Line 1 – delete "extreme", and "lightweight"; lightweight is repeated again just below.

Line 2 - "whose" - typically used to refer to humans - better to say: "and has well documented and freely available code and hardware designs."

Line 4- "conventional proprietary data recording systems".

Line 6 – remove dash in data-logger to be consistent. Also check through document for consistent writing of data logger.

Line 11 – First refer to Figure 1, before referring to Figure 2.

Line 17 – Better: "entry barrier".

Line 23 – EAGLE – add reference with link to program.

Line 28 – Mention what doxygen is.

Line 31 – "The ALog series comprises three main data loggers". Mention this sentence earlier (abstract, introduction, and/or start of Methods description). Otherwise it is not clear why sometimes it is talked about the ALog data logger (singular) and sometimes the data logger series (plural). This could also help with referring to Figure 1 before Figure 2.

Line 31/32 – list the ALog data loggers in same sequence as in figure, i.e. first ALog Shield 2.2. Also, better to first discuss the ALog Shield 2.2, then follow with the BottleLoggers, to avoid confusion between data loggers.

Figure Caption – ensure consistent naming of the three data loggers. E.g., (a) The ALog Shield 2.2

Line 1 – What are these performance upgrades? Please elaborate more.

Line 6 - First the ALog Shield 2.2 is described in detail, but then no further details are provided right away for the following models. I assume much of what comes below refers mostly to the later versions, this should be made clear however through a transition sentence.

Line 7 – Each of the ALog data loggers? Better: Each data logger of the ALog series contains. . .

Line 9 – Has this been tested in the field? See comments above too, and add reference to field experience here, or later on in field section.

Line 11 - SD cards are also easy to download data from for field assistants / citizen scientists who are not technical experts.

Line 1 – Rephrase "While a simple design decision", maybe: "While it is a simple design, using an SD card. . ."

Line 7 – "aggressive sleep cycle" - It should be explained what is meant by 'sleep cycle'. Otherwise readers who are not familiar with Arduinos might not understand. It is explained further below, consider moving the section upward, or referring to it here. Does the 'aggressive sleep cycle' refer to the 1sec per minute awake cycle?

Line 9 – remove 'an'

Line 3 and line 8 – are these values theoretical, tested in the lab, or actually experienced in the field?

Line 16 – Remove dash.

Figure 3 – darken shading for low power operations box, not clearly visible on screen.

Line 7 –mention in introductory overview or abstract that field deployment is also discussed, and examples are provided.

Line 8 to 12 – could mention some examples of deployment in abstract, this really strengthens the ALog argument.

Line 25 – remove dash

Line 26/27 - Delete this last sentence, this is not relevant to the ALog development and out of context here. Instead, you can highlight the consistent data recording over the period of a year in extreme conditions.

Line 30 – reference to Figure 5b should come at end of sentence, as, if I understand correctly, both sensors are within the solar radiation shield visible in the Figure?

Figure caption (e) – Better to remove 'covary' here, and highlight consistency of recording instead. Add start and end date of recording. Refer to paper where these data are discussed.

Line 9 – ideally. . .

Line 10 - Could make this point earlier on in introduction already, i.e. that Arduinos were originally developed/are often used for hobby electronics etc by the general public.

Line 11 – delete 'movement'

Line 23 – "In doing so"

Line 11 – user guides

Line 12 – Supplemental

Line 12 – Add University, location and page numbers, also for other theses that are cited.

Line 16 – add publisher and page numbers, link etc.

Line 2 - consistency - the other AGU reference included location and date. Include this here.

Line 6 – add reference link

Line 10 – add publisher information.

---

## Author Comment (AC1) · 19 Feb 2019

We thank Dr. ir. Hut for his multiple constructive comments on our manuscript. We plan to make multiple changes and improvements in response to his comments. We note these changes in the responses to his enumerated points (below; original text not repeated):

1. Telemetry is increasingly included in field data-logging systems, so this is a fair and important point. We plan to add the following text to a revised manuscript [see Discussions paper for bibliographic entries]:

- "(i.e., no built-in telemetry)" – after "standalone units"

[Figure]

- After the paragraph on power consumption, we have added this paragraph: "As a result of our desire to minimize power consumption, which is especially important for field deployments in remote regions, we decided not to include on-board telemetry. Off-board radio (e.g., RFM95, XBee), mobile phone (e.g., Particle Electron, Particle Boron), or satellite (SPOT, Iridium) telemetry packages could be added through the exposed digital interfaces on the ALog data logger. However, such additions would require their own significant power paths, including rechargeable batteries, charge controllers, and solar panels, thus negating much of the low-power benefit of the ALog BottleLogger design. Other designs – including the MayFly data logger (Hicks et al., 2015), which includes an XBee header and firmware support for radio telemetry (Aufdenkampe et al., 2017), and direct logging by Adafruit Feather or Particle internet-of-Things ("IoT") boards, so long as data can be telemetered and timestamped rapidly enough that the lack of an accurate on-board real-time clock is not a problem – are good options where data return and not power consumption is the variable to optimize."

2. We will add a new subsection on "Enclosures" inside the "Field deployment" section. We plan for its text to read:

"Choosing an appropriate enclosure is a key decision for equipment survival in the field. The "BottleLogger" moniker comes from its designed form factor that allows it to fit inside a wide-mouth Nalgene bottle. Such bottles seal well and are commonly available from suppliers of both laboratory and outdoor equipment. This design feature was created as an option when easier-to-use but harder-to-souce enclosures are not available. In the majority of our deployments, we have used ABS NEMA (i.e., outdoor-rated) enclosures (models NBF-32104 and NBF-32108). These boxes are gasketed, include lever-style clips for easy opening and closing in the field, are large enough for either 3×D or 3×AA cells, and may be easily drilled or machined to accommodate cable glands for connections to sensors. The shorter enclosure (NBF-32104) requires

Interactive
comment

a right-angle barrel jack plug in order to fit the length of the logger, SD card, and power connector into the box.

We typically attach the loggers to the lid of the enclosure and the battery pack to the bottom of the enclosure using self-adhesive hook-and-loop. This holds both in place, but allows either to be easily removed for wiring. The longer boxes (NBF-32108) permit cable glands to be drilled in the lid next to the logger, reducing the need for cable strain-relief. The shorter boxes (NBF-32104) may include cable glands in one or more sides, if fitted with a $3\times$AA cell pack."

3a. Releases on GitHub for all relevant repositories were made prior to release.

3b. These releases were indexed on Zenodo prior to publication and are available in the "Assets" tab on the article page. However, I realize from Dr. ir. Hut's comments that I have neglected to include them in the main text. As EGU journals now require this (as is sensible), We will add references for all seven hardware, firmware, and software assets and cite them at appropriate places throughout the article.

4. This is a good point, and I will go farther to note that we did not explicitly describe interval-based logging (using the RTC) in any real detail. To remedy this, we will add the following text after the paragraph that begins "The sensors component includes...":

"Sensors may be read on a standard interval or in response to an event. When reading measurements at a standard interval (typically 1–10 minutes), the RTC wakes the ALog using an interrupt. Once awake, the ALog retrieves data from all sensors recording environmental states. Reading and recording data from these sensors typically takes 1–3 seconds, during which the ALog is operating in its high-power "awake" state (Table ??). An event-based impulse, such as that from a tipping-bucket rain gauge, instantaneously wakes the ALog and records a time stamp to a different data file from that which is used for regular RTC-driven measurements. Reading and recording this

time stamp typically requires <0.3 s of awake-state power consumption. If the ALog is already awake (e.g., during RTC-driven data logging) when an event occurs, the ALog firmware records the time of the event to its file and then continues the remainder of its ongoing task."

5. Not having a clear cost breakdown is a major omission on our part! We have updated our bills of materials and will add in a table in which we specify components, PCB, and labor costs for different quantities.

---

## Author Comment (AC2) · 19 Feb 2019

We thank the anonymous referee for their kind words and detailed constructive criticism in a project that has been a longstanding labor of love. The referee's comments are in Roman text, whereas our responses are *italicized*.

General comments:

This article describes the development and technical details of the ALog data logger series, an open-source and low cost data logger that is based on Arduino technology. The article and the described data loggers are a significant contribution to the science community and readers of HESS, as the data loggers may provide a useful technology

to many environmental scientists. The article summarizes the substantial development that has gone into the data logger development over many years, and provides detailed background information. The article also includes supplemental material and codes provided online. This helps to make the data logger accessible to the science (and general public) community. The article is well written and well organized throughout, with clear descriptions of the technology. I only have a few minor questions, mostly regarding field deployment, and minor comments that should be addressed. Apart from these, I recommend this article for publication.

*Thank you.*

Specific Comments:

1) Please also discuss some of the challenges that you have faced with the ALog data logger in the field, and that a potential user of this data logger may encounter and should be aware of. You mention several field experiments with ALog data loggers in adverse conditions. How long were the data loggers actually in the field, how robust were they found to be? What field issues did you encounter that were specific to the ALog? Were you able to remidify these in the next iterations? I realize the ALogs have been developed over a long period, but a few more examples would be helpful to a potential user of the technology.

*We will add a sentence stating, "Field deployments ranged from a few days to three years.". Furthermore, we will add a full paragraph on field deployments to demonstrate how we have developed in response to challenges/failures.*

2) For instance, did you encounter clock drift? It is referred to a really low clock drift value in the article, but was this value based on 'theoretical lab experiments', or tested in the field?

*We encountered minimal clock drift in the field, and insofar as we were able to tell, the*

[Figure]

*clock remained in spec. Because this was not tested by us rigorously (i.e., these are just our casual observations) and the reported drift is simply the data-sheet-provided value for the full temperature range, we do not wish to comment very extensively on its accuracy. However, this is a very common component from a reputable manufacturer.*

3) The low power use is impressive. Was this value also experienced in the field? I.e., one data logger actually ran for ∼2 years on three AA batteries? Or is this a theoretical value based on consumption? Also, what kind of sleep-awake cycling is typically used? A 1 second per minute interval is mentioned, was that typically used? It probably depends on sensor and application, but some examples would be good.

*We have run loggers in the field for >1 year on alkaline batteries (typically D), but we try not to let the batteries die completely! We will add text to clarify that these are calculations extrapolated from lab measurements with partial field validation. Towards the question about the sleep cycle, we will add a sentence in the paper stating, "In our field deployments, we typically recorded data once every ten minutes, further increasing battery life."*

4) In line with the above questions, please also include some more information on how the data loggers were installed in the field. What kind of encasing have you found to work well with these data loggers? Do you typically use batteries or solar panels?

*We will add a section on enclosures based both on this comment and one by the other referee (Hut). We will add text indicating that we practically always used batteries due to the low power consumption.*

Technical corrections

*If a specific technical correction is not listed here, it is because we plan to correct it precisely as suggested by the referee and therefore had no comment to make.*

P. 1, Line 17 – likely also capacity challenges, especially in developing countries.

*We will add, "technology that can function and be repaired in least-developed countries (Reda et al., 2017), ..."*

P. 2, Line 1 – delete "extreme", and "lightweight"; lightweight is repeated again just below.

*Text will be updated to: "What the field-monitoring community requires from the open-source movement is a low-power, modular, single-board data logger that is easy to use and whose code and hardware designs are well documented and freely available."*

P. 2, Line 2 – Line 2 - "whose" - typically used to refer to humans - better to say: "and has well documented and freely available code and hardware designs."

*Understood, but "whose" is actually correct in this case, see https://www.merriam-webster.com/words-at-play/whose-used-for-inanimate-objects*

P. 2, Line 6 – remove dash in data-logger to be consistent. Also check through document for consistent writing of data logger.

*This is a hyphen rather than a dash, and is required when two nouns are used together to modify another noun. When "data logger" is not used in this way, there should be no hyphen. I will check to make sure that this is the case.*

P. 3, Line 1 – What are these performance upgrades? Please elaborate more.

*Will update text to, "version 3.0 has a more powerful microcontroller core and a dedi-*

*cated 16-bit analog-to-digital converter (ADC)"*

P. 3, Line 6 – First the ALog Shield 2.2 is described in detail, but then no further details are provided right away for the following models. I assume much of what comes below refers mostly to the later versions, this should be made clear however through a transition sentence.

*I will add a sentence stating that, "Both are described in more detail in this section."*

P. 3, Line 9 – Has this been tested in the field? See comments above too, and add reference to field experience here, or later on in field section.

*I will add the phrase here: "based both on extrapolation from laboratory power-consumption measurements (Table 1) and field deployments (Armstrong et al., 2016)"*

P. 3, Line 11 – SD cards are also easy to download data from for field assistants / citizen scientists who are not technical experts.

*Thank you for this point. We will add, "The use of text files on SD cards also simplifies the act of downloading and viewing the data, making it easier for field staff and citizen scientists to work with the ALog." We will add this to the paragraph after the one in which this suggestion was made, as this is entirely about the SD cards.*

P. 4, Line 1 – Rephrase "While a simple design decision", maybe: "While it is a simple design, using an SD card. . ."

*"While it is a simple design decision, using an SD card..."*

P. 4, Line 7 - "aggressive sleep cycle" - It should be explained what is meant by 'sleep

cycle'. Otherwise readers who are not familiar with Arduinos might not understand. It is explained further below, consider moving the section upward, or referring to it here. Does the 'aggressive sleep cycle' refer to the 1sec per minute awake cycle?

*We will make this more descriptive: 'we implemented a "sleep" cycle to shut down all non-essential subsystems while not logging'*

P. 6, Line 3 and line 8 – are these values theoretical, tested in the lab, or actually experienced in the field?

*We will clarify that these are laboratory tests; we never allowed the batteries to fully run down in the field: we needed our data!*

P. 6, Line 16) – Remove dash. *(and later instance)*

*This hyphen is grammatically required. For a quick review, see https://www.grammarly. com/blog/hyphen-with-compound-modifiers/*

P. 9, Line 7 – mention in introductory overview or abstract that field deployment is also discussed, and examples are provided.

*Will add to abstract: "The ALog has been deployed at field sites in Colorado, Alaska, Louisiana, and Minnesota, USA; Ontario, Canada; Argentina; and Ecuador." For the introductory section, we retain, "We iterated development and field testing from 2010 to present"*

P. 9, Line 8 to 12 – could mention some examples of deployment in abstract, this really strengthens the ALog argument.

*Thank you for this suggestion; our response to your above comment is based on these lines.*

P. 9, Line 26/27 – Delete this last sentence, this is not relevant to the ALog development and out of context here. Instead, you can highlight the consistent data recording over the period of a year in extreme conditions.

*We will replace this sentence with: "Figure 4e contains the first five days of data from this deployment, which lasted one year."*

P. 10, Figure caption (e) – Better to remove 'covary' here, and highlight consistency of record- ing instead. Add start and end date of recording. Refer to paper where these data are discussed.

*I think that the figure itself demonstrates the consistency of the data, and it also indi- cates the dates of the recordings. I now note in the main text that the station recorded for one year. I feel it might be valuable to include a bit of interpretation, as an example of a use case. These data are published only in the present work.*

P. 11, Line 10 – Could make this point earlier on in introduction already, i.e. that Ar- duinos were originally developed/are often used for hobby electronics etc by the gen- eral public.

*We will add a brief mention of this in the short introduction (to give it appropriate weight) as follows: "Hardware advances alone cannot produce an effective standalone mea- surement platform, so we paired our new designs with custom-built firmware libraries – built atop the popular and easy-to-use Arduino platform – and software to streamline data-logger programming." (Text between the endashes is newly proposed.)*

P. 11, Line 23 – "In doing so"

*"In so doing" is the intended phrasing.*

P. 13, Line 11 – user guides

*We will remove this phrasing and instead refer to these via their doi and reference, following the recommendation of R. Hut and, indeed, EGU policy on citation of data/code/etc. items.*

P. 13, Line 12 – Supplemental

*"Supplementary" is used by EGU journals, e.g., https:// www. hydrology-and-earth-system-sciences.net/ for_authors/ submit_your_manuscript.html*

P. 14, Line 12 – Add University, location and page numbers, also for other theses that are cited.

*University is included. Strangely enough, EGU journals do not seem to include page numbers (at least not per their BibTeX style file).*